# A SCALABLE COOPERATIVE/COMPETITIVE SPLITTING SCHEME FOR MIXTURE OF EXPERTS MODELS

## ABSTRACT

We present a novel probabilistic expectation-maximization scheme for training hierarchical mixture-of-experts models that both exposes and exploits parallelism during training. By replacing the typical categorical distribution used in gating networks with a joint distribution blending cooperative and competitive mechanisms, we obtain a likelihood that encodes both global and local interactions between experts. The application of an M-splitting scheme reveals an M-step that enables the solution of localized, embarrassingly parallel subproblems governing local experts, with deferred corrections accounting for global coupling between experts. When combined with a hierarchical decomposition of nested networks, this yields a fast multi-level training scheme reminiscent of multigrid algorithms, which avoids under-utilization of experts, exposes further GPU parallelism and outperforms standard models on regression tasks. We provide experiments using a scalable GPU implementation that demonstrate rapid convergence and parallel scalability of the iterative scheme, as well as strong localization of the model for non-smooth, high-dimensional regression problems.

## 1 INTRODUCTION AND PRIOR WORK

Mixture-of-experts (MoE) models employ a gating network to dynamically route specialized sub-networks ("experts") to perform a given task. When the gating network is sparsely activated, the resulting architecture can be highly efficient, as only a small number of experts are evaluated per input. Following their introduction in the 1990s Jacobs et al. (1991); Jordan and Jacobs (1994), MoE models were initially viewed as a means of incorporating modular, specialized components that performed well across multiple tasks French (1999). In recent years, however, they have attracted attention as a scalable approach to training large models Shazeer et al. (2017), with some works successfully training models with over 100 billion parameters Fedus et al. (2022); Du et al. (2022); Lewis et al. (2021); Lepikhin et al. (2020). The recent success of DeepSeek-V2 also relies crucially on MoE architectures Liu et al. (2024). Whereas large monolithic architectures of comparable size are challenging to train, sparsely gated expert models offer a way to combine easier-to-train component networks.

Despite their scalability, training MoE models remains challenging, particularly with respect to ensuring balanced expert utilization and promoting cooperation among experts. Traditional training methods often struggle with these issues, leading to underutilized experts and suboptimal performance.

In this work, we construct a probabilistic framework that admits a scalable expectation-maximization (EM) training scheme for efficiently training individual experts. We introduce a new approach that combines the predictive accuracy of *cooperative* experts, which blend expert predictions, with the computational efficiency of *competitive* experts, which train independently on disjoint subsets of data. This approach integrates several novel technical contributions:

- A novel probabilistic gating function exposes a decomposition of local and global work by using a mean-field assumption to blend cooperative and competitive effects.
- An expectation-maximization (EM) algorithm, solved via a novel splitting scheme, exposes parallelism in local computations amenable to GPU acceleration while preserving cooperation across experts. A theoretical bound establishes a sufficient condition for convergence which are explored experimentally.

- A hierarchical MoE architecture supports a novel multi-level training scheme that ensures full utilization of all experts. This work builds upon the multilevel training framework introduced in Trask et al. (2022) (see Appendix A for distinctions). While the current method inherits this structure, it introduces a cooperative/competitive split that exposes latent GPU-parallelism.

- Benchmarks demonstrate acceleration from these techniques, as well as the importance of cooperative effects in a physics-informed architecture where a purely competitive expert model provides over $10\times$ less accurate predictions.

For simplicity, we develop the method and accompanying theory in the context of supervised regression using simple dense feedforward expert models. However, the proposed scheme is readily compatible with switch transformers and can be integrated into large-scale models without modification. The benchmarks presented in this work demonstrate strong GPU scalability, establishing the feasibility of extending the approach to large transformer-based architectures in future work.

**Probabilistic MoE, EM, and cooperative/competitive experts.** Early work by Jordan and Jacobs (1994) demonstrated that MoE models admit a probabilistic interpretation, with the gating network defining a categorical distribution upon which experts are conditioned. By maximizing the evidence lower bound (ELBO), they showed that EM can be used to efficiently train experts *competitively*: the M-step yields posterior-weighted, decoupled subproblems for training each expert independently. They also proposed a *cooperative* loss that fully couples experts during training, improving accuracy at the cost of increased computational expense. More recent works have explored the trade-off between cooperative and competitive training objectives Ahn and Sentis (2021); Do et al. (2025). The present work provides a probabilistic unification of both losses under a single ELBO formulation.

**Hierarchical models and multigrid.** Our cooperative/competitive split builds on the hierarchical MoE regression framework of Trask et al. (2022), which extends Jordan and Jacobs (1994) by incorporating multigrid-inspired training. In numerical linear algebra, multigrid methods enable $O(N \log N)$ solves Brandt (1977); Briggs et al. (2000) and power exascale simulations Ibeid et al. (2020); Falgout et al. (2021). This motivates our hypothesis that multigrid-trained hierarchical models can scale to similarly large architectures. Recent work has explored multigrid-inspired model training Ke et al. (2017); Albergo et al. (2019); Gunther et al. (2020) and ML-enhanced multigrid solvers Oswald et al. (2023); Taghibakhshi et al. (2023).

**Splitting schemes in scientific computing.** Splitting schemes are an essential algorithmic ingredient in scientific computing, with seminal works by Chorin (1968) and Strang (1968). Our work leverages splitting methods in numerical linear algebra, like Jacobi and Gauss-Seidel, that were among the first iterative methods for solving linear systems (Saad, 2003; Golub and Van Loan, 2013). Within multigrid methods, they serve a critical numerical role of reducing high-frequency errors (Brandt, 1986). Further, exploiting the parallelization of splitting schemes, like Jacobi, allows multigrid methods to run efficiently on supercomputers Adams et al. (2003); Chow et al. (2006).

**Scientific machine learning.** Beyond scaling large models, MoEs have seen growing use in ML-based physical modeling—e.g., in fluids Sharma and Shankar (2024); Zigon and Zhu (2025), chemistry Shirasuna et al. (2024), and materials Chang et al. (2022). They are particularly effective for enforcing hard constraints on physics residuals Chalapathi et al. (2024); Actor et al. (2024), where accuracy approaching machine precision are often required. While some address this via problem-specific multistage training Wang and Lai (2024); Ainsworth and Dong (2021); Howard et al. (2023), our method aims to achieve such accuracies directly through optimizer design. We demonstrate this by solving a physics-informed neural network with hierarchical cooperative training—without modifying the PDE—yielding accurate predictions. This scales the earlier findings of Cyr et al. (2020), which showed orders-of-magnitude improvements from cooperative MoE losses but required dense $O(N^3)$ linear solves that hinder scalability.

Code is available on the (anonymized) GitHub `https://anonymous.4open.science/r/coopcompsplit_neurips2025-3F27/readme.md`.

## 2 TECHNICAL APPROACH

We summarize here the key algorithmic features to exposing and exploiting parallelism, sketched in Figure 1. For further details and distinctions from Trask et al. (2022), see Appendix A.

MoE models admit a probabilistic interpretation where each expert prescribes a conditional distribution and the gating network defines a categorical distribution selecting a given expert. Let the gating variable $Z(x) \sim \text{Cat}(\pi(x))$, where $\pi(x)$ prescribe the probability of a given input $\boldsymbol{x}$ being assigned to the $i$-th expert model, i.e., $p(Z_1(\boldsymbol{x}) = i_1) = \pi_{i_1}(\boldsymbol{x}; \theta)$, providing the output distribution

$$p(Y_1(\boldsymbol{x}) = y) = \sum_{i_1=1}^{N_1} p(\mathcal{E}_{i_1} = y | Z_1(\boldsymbol{x}) = i_1) p(Z_1(\boldsymbol{x}) = i_1),$$

We assume a mean-field decomposition of the experts into cooperative and competitive components

$$p(\mathcal{E}_{i_1} = y | Z_1(\boldsymbol{x}) = i_1) = \frac{1}{Q_{i_1}} \mathcal{N}(y; \mu_{i_1}, \sigma_{\text{comp}}^2) \mathcal{N}(y; \hat{y}_1, \sigma_{\text{coop}}^2), \tag{1}$$

where $Q_{i_1}$ is the normalizing factor. Each expert prediction $\mu_{i_1}$ and the cooperative output blending all experts $\hat{y}_1$ are given by

$$\mu_{i_1}(\boldsymbol{x}) = \mathbf{c}_{i_1}^\top \boldsymbol{H}_{i_1}(\boldsymbol{x}; \theta) \quad \text{and} \quad \hat{y}_1(\boldsymbol{x}) = \mathbb{E}[Y_1](x) = \sum_{i_1=1}^{N_1} \pi_i(\boldsymbol{x}; \theta) \mu_i(\boldsymbol{x}),$$

respectively. The assumed expert form is consistent with a generic hidden architecture consisting of a hidden layer $\boldsymbol{H}_{i_1}$ composed with a linear layer $\mathbf{c}_{i_1} \in \mathbb{R}^{N_{\text{basis}}}$. The gating distribution $\pi_{i_1}(\boldsymbol{x}; \theta)$ is parameterized by a neural network with a softmax activation at the output layer. This setting encapsulates a broad range of architectures, including switch transformers; for simplicity, in this work, we will consider ResNet architectures.

In general, this architecture requires training of individual experts to prescribe $\mu_{i_1}$, specification of additive noise $\sigma_{\text{comp}}^2$ and $\sigma_{\text{coop}}^2$, and training of the gating network. We will demonstrate a novel expectation maximization strategy that provides a decoupling of the training for individual experts; specifically, we obtain a coupled system of equations for optimal linear layer weights $\mathbf{c}_{i_1}$. To train these individually, we may reinterpret $\sigma_{\text{comp}}^2$ and $\sigma_{\text{coop}}^2$ instead as numerical parameters that may be used to control the relative importance of local and global information, selecting the cooperative contribution to be non-zero but sufficiently small that guarantees can be provided for a splitting scheme.

**Hierarchical generalization to arbitrary levels.** We extend to a multilevel setting by introducing a hierarchy of latent variables $Z_n$ for each level $n$ defined conditionally on the previous $n-1$ levels

$$p(Z_n = i_n \mid Z_1 = i_1, \cdots, Z_{n-1} = i_{n-1}) = \pi_{\text{I}_n},$$

$$p(Z_1 = i_1, \cdots, Z_n = i_n) = \prod_{k=1}^{n} \pi_{\text{I}_k} = \tilde{\boldsymbol{\pi}}_{\text{I}_n},$$

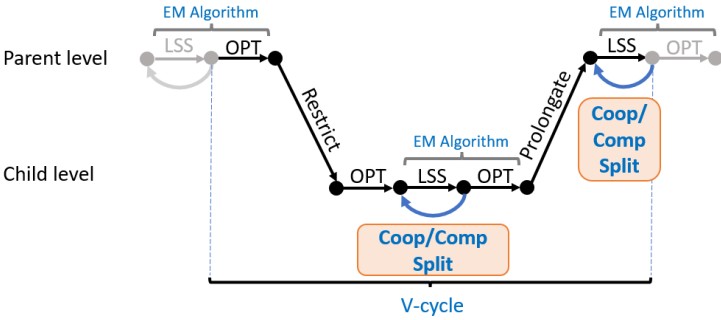

Figure 1: An illustration of the hierarchical training. Details are provided in Appendix A.

$$p(Y_n(\boldsymbol{x}) = y) = \sum_{\mathrm{I}_n} \tilde{\boldsymbol{\pi}}_{\mathrm{I}_n} \mathcal{N}(y; \mu_{\mathrm{I}_n}, \sigma_{\mathrm{comp}}^2) \mathcal{N}(y; \hat{y}_n, \sigma_{\mathrm{coop}}^2),$$

$$\mu_{\mathrm{I}_n}(\boldsymbol{x}) = \mathbf{c}_{\mathrm{I}_n}^\top \boldsymbol{H}_{\mathrm{I}_n}(\boldsymbol{x}; \theta), \quad \hat{y}_n(\boldsymbol{x}) = \sum_{\mathrm{I}_n} \tilde{\boldsymbol{\pi}}_{\mathrm{I}_n}(\boldsymbol{x}; \theta) \mu_{\mathrm{I}_n}(\boldsymbol{x}).$$

where $\mathrm{I}_n = \{i_1, \cdots, i_n\}$ is used as a shorthand for all indices up to the current level. In Trask et al. (2022), the authors demonstrate a multi-grid inspired scheme where expectation maximization is used to perform polynomial regression. In the current work, we modify this both by considering arbitrarily deep architectures for experts and using the mean-field distribution in Equation 1. Following Trask et al. (2022), at training time we perform a V-cycle optimization evaluating an EM-step at each level of the hierarchy progressing from coarse to fine, and then applying marginalization of the probabilistic to traverse the hierarchy back from fine to coarse. Details of this and a diagram of training are provided in Appendix A.

When developing a multigrid scheme, a common requirement is that the range of the coarse space is a subset of the fine space, implying that finer scales are well-approximated on coarse-scales. Formally,

$$\mathrm{span}(\boldsymbol{H}_{i_1, \cdots, i_n}) \subseteq \mathrm{span}(\boldsymbol{H}_{i_1, \cdots, i_n, i_{n+1}}). \tag{2}$$

We design architectures that achieve this by constructing children of parent experts which consist of their parents hidden layer stacked with a new hidden layer, so that

$$\mu_{\mathrm{I}_{n+1}}(\boldsymbol{x}) = \mathbf{c}_{\mathrm{I}_{n+1}}^\top \begin{bmatrix} \boldsymbol{H}_{\mathrm{I}_n}(\boldsymbol{x}; \theta) \\ \tilde{\boldsymbol{H}}_{n+1}(\boldsymbol{x}; \theta) \end{bmatrix},$$

where $\tilde{\boldsymbol{H}}_{n+1}(\boldsymbol{x}; \theta)$ is a new hidden layer architecture. This construction provides experts of increasing nested complexity as the hierarchy is extended more deeply and exposes parallelism.

**Cooperative/competitive EM update.** We next demonstrate how the assumed mean-field approximation impacts the standard EM update from Jordan and Jacobs (1994). The observed data log likelihood is given by

$$\log L(\theta; \mathcal{D}) = \sum_{d=1}^{N_d} \log \left[ \sum_{\mathrm{I}_n} \tilde{\boldsymbol{\pi}}_{\mathrm{I}_n}(\boldsymbol{x}^d; \theta) \mathcal{N}(y^d; \mu_{\mathrm{I}_n}(\boldsymbol{x}^d), \sigma_{\mathrm{comp}}^2) \mathcal{N}(y^d; \hat{y}_n(\boldsymbol{x}^d), \sigma_{\mathrm{coop}}^2) \right],$$

which, by Jensen's inequality, is bounded from below by

$$\ell(\theta) = \sum_{d=1}^{} \sum_{\mathrm{I}_n} w_{\mathrm{I}_n}(\boldsymbol{x}^d) \log \frac{\tilde{\boldsymbol{\pi}}_{\mathrm{I}_n}(\boldsymbol{x}^d; \theta) \mathcal{N}(y^d; \mu_{\mathrm{I}_n}(\boldsymbol{x}^d), \sigma_{\mathrm{comp}}^2) \mathcal{N}(y^d; \hat{y}_n(\boldsymbol{x}^d), \sigma_{\mathrm{coop}}^2)}{w_{\mathrm{I}_n}(\boldsymbol{x}^d)}.$$

We choose $w_{\mathrm{I}_n}(\boldsymbol{x}^d)$ such that the ELBO is a tight lower bound,

$$w_{\mathrm{I}_n}(\boldsymbol{x}^d) := p(Z_1 = i_1, \cdots, Z_n = i_n \mid Y_n = y^d) = \frac{\tilde{\boldsymbol{\pi}}_{\mathrm{I}_n} \mathcal{N}(y^d; \mu_{\mathrm{I}_n}(\boldsymbol{x}^d), \sigma_{\mathrm{comp}}^2)}{\sum_{\mathrm{J}_n} \tilde{\boldsymbol{\pi}}_{\mathrm{J}_n} \mathcal{N}(y^d; \mu_{\mathrm{J}_n}(\boldsymbol{x}^d), \sigma_{\mathrm{comp}}^2)}.$$

Computing $w_{\mathrm{I}_n}(\boldsymbol{x}^d)$ prescribes the E-step of each iteration. In the M-step, we find the optimal parameters $\mathbf{c}_{\mathrm{I}_n}$, $\boldsymbol{H}_{\mathrm{I}_n}$ and $\tilde{\boldsymbol{\pi}}_{\mathrm{I}_n}$ to maximize the ELBO. Taking the derivative of the ELBO with respect to the expert coefficients $\mathbf{c}_{\mathrm{I}_n}$ yields the weighted least-squares problem

$$\sum_{d, \mathrm{J}_n, \beta} \left( \frac{w_{\mathrm{I}_n}^d \delta_{\mathrm{I}_n \mathrm{J}_n}}{\sigma_{\mathrm{comp}}^2} + \frac{\tilde{\boldsymbol{\pi}}_{\mathrm{I}_n}^d \tilde{\boldsymbol{\pi}}_{\mathrm{J}_n}^d}{\sigma_{\mathrm{coop}}^2} \right) H_{\mathrm{I}_n, \alpha}^d H_{\mathrm{J}_n, \beta}^d c_{\mathrm{J}_n, \beta} = \sum_d \left( \frac{w_{\mathrm{I}_n}^d}{\sigma_{\mathrm{comp}}^2} + \frac{\tilde{\boldsymbol{\pi}}_{\mathrm{I}_n}^d}{\sigma_{\mathrm{coop}}^2} \right) y^d H_{\mathrm{I}_n, \alpha}^d \qquad \text{(LS)}$$

where $\delta_{\mathrm{I}_n \mathrm{J}_n}$, a generalization of the Kronecker delta, equals 1 if and only if all corresponding components of $\mathrm{I}_n$ and $\mathrm{J}_n$ are the same, and equals 0 otherwise. Solving this weighted least-square problem requires a dense matrix solve; in the following section we demonstrate how this can be scalably solved.

$\boldsymbol{H}_{\mathrm{I}_n}$ and $\tilde{\boldsymbol{\pi}}_{\mathrm{I}_n}$ are optimized with a gradient descent loss

$$\mathcal{L}_{\mathrm{GD}} = -\ell(\theta)$$

By marginalizing the posterior of the $n$ level, we can obtain another estimator for the $n-1$ level, i.e.,

$$\hat{w}_{\mathrm{I}_{n-1}} = \sum_{i_n} w_{\mathrm{I}_n}.$$

Note that the summation is performed only on the last index $i_n$. Thus, we obtain a hierarchical sequence of M step solves traversing up the hierarchy, accessing information at each level from the solve of the previous level. See Appendix A for a worked two-level example.

## 3 Splitting scheme for parallelizable iterative solver

We can split the tensor on the left-hand side of equation equation LS into the competitive component tensor $\mathbf{M}$ and the cooperative component tensor $\mathbf{N}$ defined by

$$\mathbf{M_{IJ}} = \sum_d \frac{w_{\mathrm{I}_n}^d \, \delta_{\mathrm{I}_n \mathrm{J}_n}}{\sigma_{\mathrm{comp}}^2} H_{\mathrm{I}_n,\alpha}^d H_{\mathrm{J}_n,\beta}^d,$$

$$\mathbf{N_{IJ}} = -\sum_d \frac{\tilde{\boldsymbol{\pi}}_{\mathrm{I}_n}^d \tilde{\boldsymbol{\pi}}_{\mathrm{J}_n}^d}{\sigma_{\mathrm{coop}}^2} H_{\mathrm{I}_n,\alpha}^d H_{\mathrm{J}_n,\beta}^d,$$

where $\mathbf{I} = \{\mathrm{I}_n, \alpha\}$ and $\mathbf{J} = \{\mathrm{J}_n, \beta\}$, allowing us to matricize these tensors and to vectorize $\mathbf{c}$ by flattening each vector $\mathbf{I}$ and $\mathbf{J}$ into a single index. While $\mathbf{N}$ is a dense matrix, $\delta_{\mathrm{I}_n \mathrm{J}_n}$ makes $\mathbf{M}$ block-diagonal, which allows the computation of its inverse to be parallelized efficiently. Exploiting this structure, we replace the direct solution of the linear system with an iterative scheme of the form

$$\mathbf{Mc}^{(k+1)} = \mathbf{Nc}^{(k)} + \mathbf{b}. \tag{I}$$

**Theorem 1.** *Assume that the experts are linearly independent with respect to the $\mathbf{M}$ and $\mathbf{N}$-weighted inner-products, i.e. for $\mathrm{I}_n \neq \mathrm{J}_n$,*

$$\det\left(\frac{\tilde{\boldsymbol{\pi}}_{\mathrm{I}_n}^d \tilde{\boldsymbol{\pi}}_{\mathrm{J}_n}^d}{\sigma_{coop}^2} H_{\mathrm{I}_n,\alpha}^d H_{\mathrm{J}_n}^d\right) \neq 0, \qquad \det\left(\sum_d \frac{w_{\mathrm{I}_n}^d \, \delta_{\mathrm{I}_n \mathrm{J}_n}}{\sigma_{comp}^2} H_{\mathrm{I}_n,\alpha}^d H_{\mathrm{J}_n,\beta}^d\right) \neq 0,$$

*then the spectral radius $\rho(\mathbf{M}^{-1}\mathbf{N}) < 1$ provides a sufficient condition for convergence. Under these conditions, choosing $R_\sigma = \sigma_{coop}^2/\sigma_{comp}^2$ as*

$$R_\sigma^* \geq \frac{\max\limits_i \sum\limits_d \tilde{\boldsymbol{\pi}}_{\mathrm{I}_n}^d}{\min\limits_i \sum\limits_d w_{\mathrm{I}_n}^d}. \tag{3}$$

*guarantees convergence of the splitting scheme.*

In Appendix B we provide a proof of this result, which follows from a generalized Rayleigh quotient analysis and the Gerschgorin circle theorem (Gerschgorin, 1931). In practice, $R_\sigma$ may be either selected adaptively following equation 3, or it may be treated as a hyperparameter to be fixed to a sufficiently large value before training.

With minor modification the Successive Over-Relaxation (SOR) method Young (1954) may be adopted to further accelerate solution. At iteration $k+1$, the update rule is modified via

$$\mathbf{c}^{(k+1)} = (1 - \omega)\mathbf{c}^{(k)} + \omega \mathbf{M}^{-1}\left(\mathbf{Nc}^{(k)} + \mathbf{b}\right),$$

where $\omega \in (0, 2)$ a tunable parameter that recovers Theorem 1 for $\omega = 1$.

## 4 Numerical experiments

In our experiments, we first present a pedagogical regression task to illustrate how the hierarchical basis enhances localization across scales. We then benchmark GPU scalability, demonstrating efficient implementation of embarrassingly parallel weighted least squares solves, as well as less

obvious parallelism via JAX's `vmap`. Next, we assess the tightness and practical impact of the bound in Theorem 1, along with an ablation study showing robustness to data dimensionality. Finally, we include a scientific machine learning example in which incorporating cooperative effects yields an order-of-magnitude improvement when solving a numerically stiff partial differential equation.

Details of training data, hyperparameters, architecture, and a link to reproducible code are provided in Appendix D.

### 4.1 ILLUSTRATIVE 1D REGRESSION EXAMPLE

In Figure 2, we regress the function $y(x) = \exp\left[-\frac{1}{2}\left(\frac{x-0.5}{0.05}\right)^2\right]$ on the unit interval. For this problem, we observe that the hierarchical experts allow for an unsupervised concentration of resolution around the relevant feature of the problem. While several works have constructed multistage function approximators by hand (see e.g. Wang and Lai (2024)), this example highlights how hierarchy, cooperative gating, and optimal EM-updates combine to provide highly accurate function approximation.

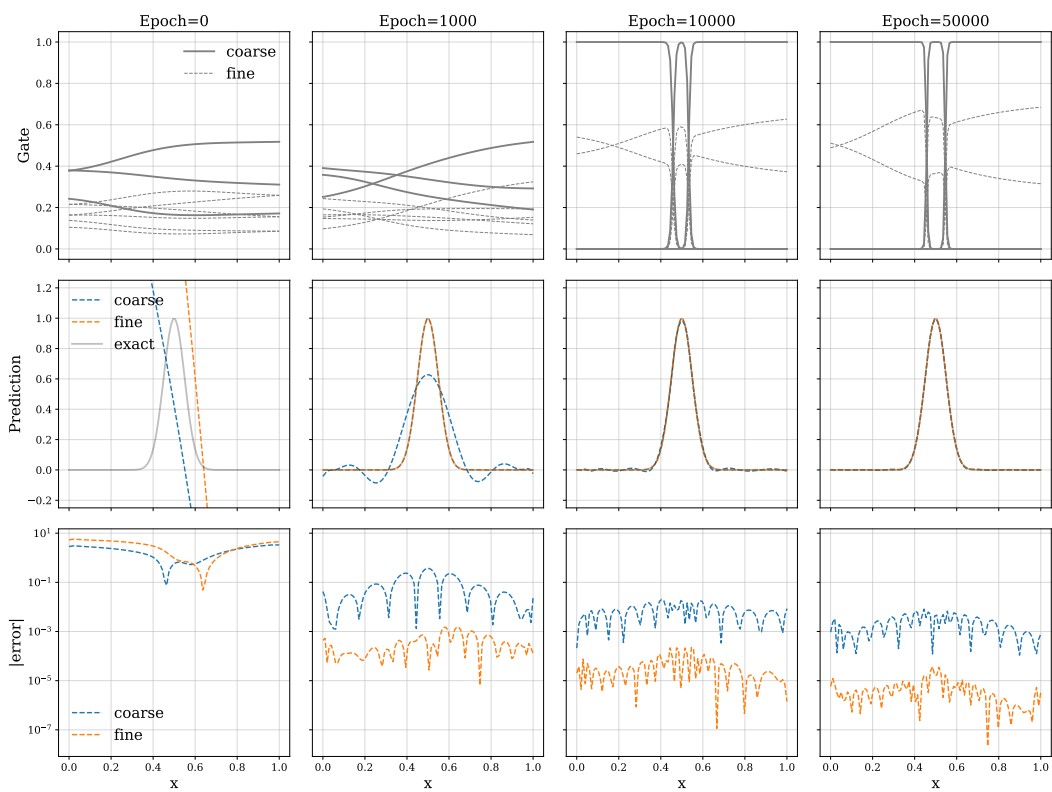

Figure 2: Regression of a Gaussian bump, highlighting unsupervised expert localization, specialization, hierarchical approximation and improved accuracy across levels. **Top:** Gating functions localize on bump, **Middle:** Upon localizing gates, experts provide refined approximation, **Bottom:** Experts deeper in hierarchy can specialize and provide orders of magnitude improved accuracy.

### 4.2 STRONG GPU SCALING OF BLENDED LEAST-SQUARES SOLVE

We consider a 64-dimensional regression problem using $N_c$ coarse and $N_f$ fine experts, benchmarking acceleration across 1, 2, and 4 GPUs relative to a single-GPU baseline. Figure 3 highlights two forms of GPU-parallelism exposed by our scheme. First, the M-step splitting decouples the global least-squares problem into many small, embarrassingly-parallel solves, yielding strong scaling. Second, the nested hierarchy exposes expert-level decoupling across levels, which JAX's `vmap` efficiently parallelizes. Appendix A details the mathematical structure behind this decoupling. Speedup is measured against a naive LU-based single-GPU solve of the monolithic system. As expected, the

split formulation achieves near-ideal scaling: $1.97\times$ and $3.79\times$ with 2 and 4 GPUs, respectively. When `vmap` is activated, we observe up to $30\times$ speedup; this improvement is directly attributable to the combination of splitting and hierarchy which allows a complete utilization of GPUs.

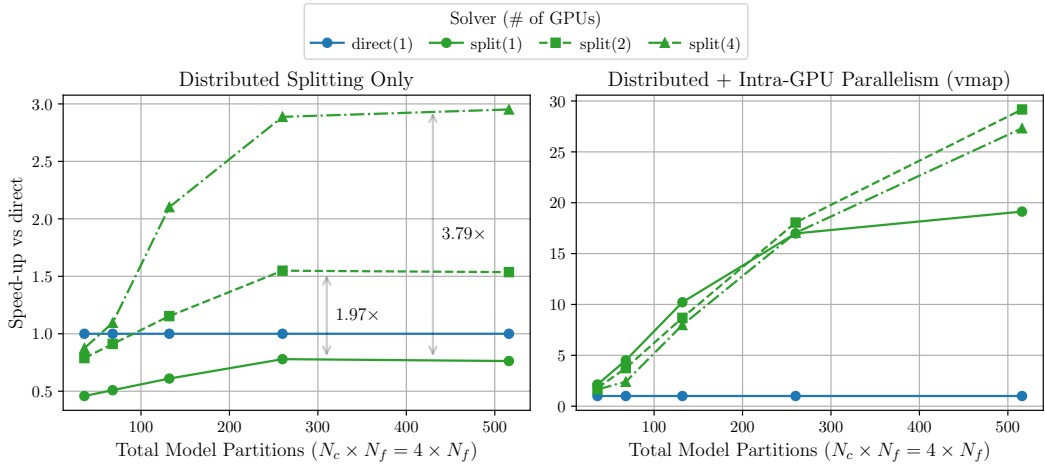

Figure 3: Strong scaling for an iteration of the split linear solver. e **Left.** Distributed splitting with no intra-GPU parallelism achieves near ideal speedup. **Right.** JAX's `vmap` exploits intra-GPU parallelism exposed by splitting and model hierarchy.

### 4.3 TIGHTNESS OF BOUND IN THEOREM 1 AND PRACTICAL IMPLICATIONS

The fundamental assumption of the method is that $R_\sigma$ may be chosen small enough to incorporate cooperative effects but large enough to guarantee convergence of the splitting. The bound for $R_\sigma^*$ in Theorem 1 makes several assumptions, raising the question of whether it is tight enough to provide practical significance. **1.** The experts are assumed linearly independent, which is particularly unlikely at initialization (see e.g. Cyr et al. (2020) for discussion of degenerate rank at initialization time), **2.** Expert posterior collapse may lead to an explosion in $R_\sigma^*$ as $w_{id} \to 0$ (See discussion in Appendix B), and **3.** the derived using Gerschgorin is too pessimistic to provide a tight and predictive estimate. We design an experiment to explore two hypotheses: **1.** There exists an experimentally calibrated constant $C$ such that choosing $R_\sigma = C\,R_\sigma^*$ gives an effective threshold for guaranteeing convergence; and **2.** at the conclusion of training the experts are linearly independent so that $R^*$ is tight.

To test, we construct an artificial scenario in which we generate several different training runs corresponding to a range of $R_\sigma$ values spanning 1 to $10^{10}$. Define $N_{split}$ as the number of iterations for the splitting scheme to converge to a given tolerance. At each epoch, we perform an ablation study fixing the matrices $A$ and $B$ and sweeping over different values of $\sigma_{coop}$ and $\sigma_{comp}$ to identify the dependence of $N_{split}$ on alternate weightings of $M$ and $N$. This provides a realistic range of possible weightings which can illustrate whether the bound in Theorem 1 is tight. We arbitrarily impose a maximum iteration count of 2000 as an indicator of divergence.

Figure 4 illustrates the results of this study. By plotting $N_{split}$ as a function of $R_\sigma/R_\sigma^*$, we obtain a scatter plot with a clear delineation consistent with selecting $C = 1e4$. Realistically $C$ may be problem dependent, but this suggest a practical strategy where $C$ may be gradually increased for a given problem until it is sufficiently large. Interestingly however, when plotting a range of values at the end of training we observe that the transition to convergence occurs at the theory-predicted $C = 1$, suggesting that the bound is tight at the conclusion of training.

### 4.4 INSENSITIVITY OF SPLITTING SCHEME TO PROBLEM DIMENSION

We consider a regression problem on the d-dimensional unit hypercube, regressing a multivariate Gaussian problem with mean at the center of the cube and standard deviation $\frac{1}{10}$. Similar to Section 4.1, this provides a test whether the gating function can localize on a simple feature in high-dimensions,

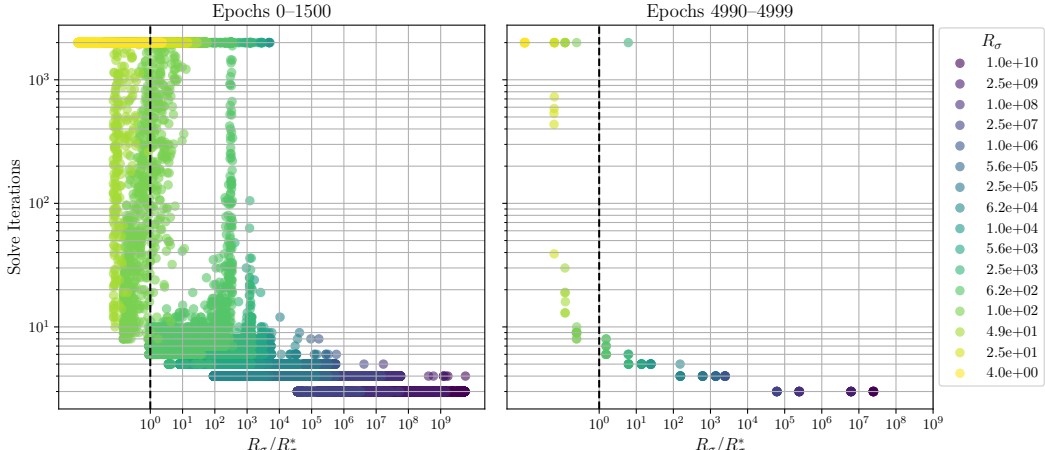

Figure 4: Ablation study exploring the tightness and practical implications of Theorem 1. **Left.** A scatter plot of the number of splitting iterations to convergence as a function of $R_\sigma/R_\sigma^2$ illustrates a sharp transition at $R_\sigma/R_\sigma^2 = 10^4$. **Right.** At the conclusion of training, we see that the predicted bound $R_\sigma = R_\sigma^*$ denoted by a vertical dashed line clearly indicates the threshold of stability, suggesting that the assumptions of Theorem 1 are valid.

and indicating whether the split cooperative/competitive scheme may realistically be deployed on high-dimensional problems.

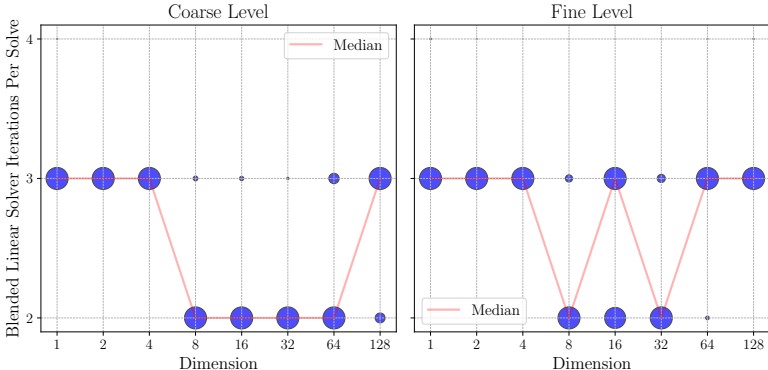

Figure 5: Ablation study illustrating insensitivity of convergence for splitting scheme to problem dimension. Radius of blue circles denotes the frequency of convergence for a given dimension for both coarse (**Left**) and fine (**Right**) levels of hierarchy. Performance is independent of dimension, illustrating suitability for high-dimensional machine learning tasks.

### 4.5 PERFORMANCE ON A PHYSICS-INFORMED NEURAL NETWORK

Physics-informed neural networks (PINNs) offer a simple case where neural networks provide candidate solutions that minimize a partial differential equation (PDE) residual Lagaris et al. (1998); Raissi et al. (2019). Many works have established pathologies in the training of PINNs Wang et al. (2021); Krishnapriyan et al. (2021); Fuks and Tchelepi (2020). While there are multiple challenges, one is that solutions to PDEs have strict regularity requirements on continuity, which many have shown may be avoided by using more advanced PDE discretizations Yu et al. (2018); Patel et al. (2022). We show in Figure 6 that an application of a cooperative expert to the original "vanilla" scheme is sufficient to achieve results without modification to the original scheme and only via choice of architecture/optimizer. We consider as a benchmark the singularly perturbed advection-diffusion equation, which in the limit as transport becomes advection-dominated, exhibits many sharp

gradients that are challenging even for mature PDE-solution techniques Roos (2008). This regularity is preserved by cooperative experts, but competitive experts are unable to resolve it. This problem requires a non-trivial extension of the framework to a multi-objective loss (See Appendix C) that provides a concrete example of how the framework may be extended to a broader class of problems.

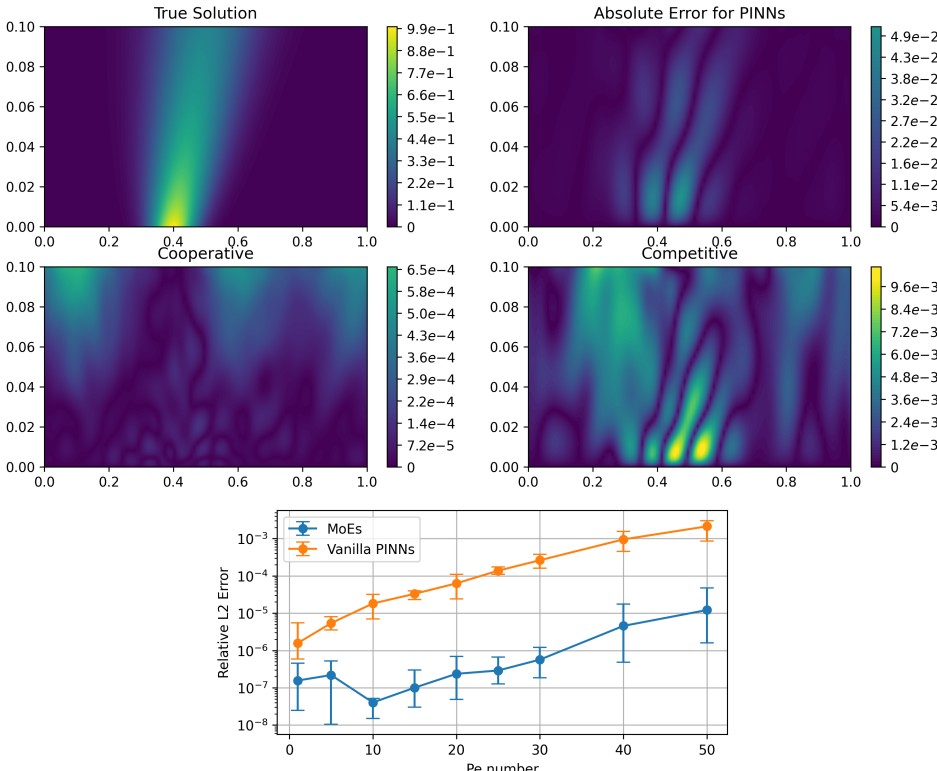

Figure 6: **Extension to multi-objective loss: Physics-informed neural networks (PINNs). Top.** In the advection-diffusion problem, High-Péclet cases yield steep gradients and pathological behavior while requiring continuity. Compared to the analytic solution (**Top-left**), standard PINNs exhibit large errors at steep gradients (**Top-right**), which are mitigated by the cooperative scheme (**Bottom-left**). Competitive experts improve training stability but yield solutions with $10\times$ larger error (**Bottom-right**). **Bottom.** An ablation over Péclet numbers in 1D shows the cooperative scheme consistently maintains $> 10\times$ lower error as the problem becomes more singular. Error bars indicate min/max error across five random seeds.

## 5 CONCLUSION AND FUTURE WORK

We introduced a novel splitting scheme for cooperative/competitive hierarchical mixture-of-experts models and provided analytical criteria guaranteeing robust performance. On simple tasks, the method exposes new forms of GPU parallelism, yielding effective results for both regression and multi-objective optimization, with performance largely insensitive to input dimension. While scaling this scheme to large transformer-based architectures remains a substantial engineering challenge, the present work establishes a clear proof of concept and a promising foundation for future development.

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

# A  DETAILS OF ALGORITHM AND DISTINCTION FROM TRASK ET AL. 2022

For specificity, we provide here the individual steps of a two-level scheme, illustrating the operations performed in a single V-cycle in Figure 7. We define operations consisting of $OPT$, $Restrict$, $LSS$ and $Prolongate$ appearing in the figure. For further derivation, we direct the interested reader to Trask et al. (2022), and provide a Github for a scalable GPU implementation at `https://anonymous.4open.science/r/coopcompsplit_neurips2025-3F27/readme.md`.

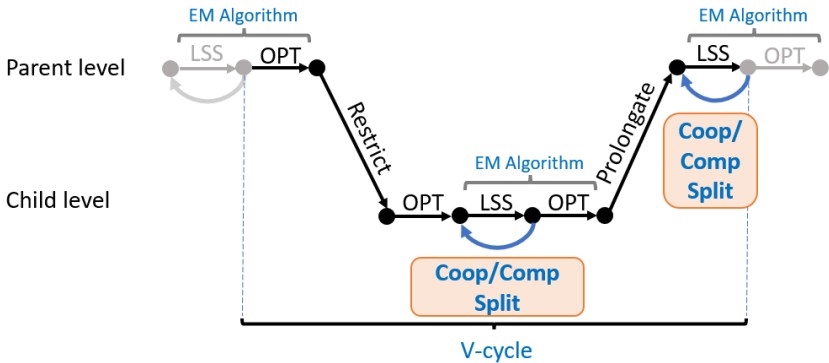

Figure 7: An illustration of the hierarchical training for a simple two level scheme. The key contribution of this work is the introduction of the cooperative/competitive splitting which allows expensive least square solves (LSS) to be distributed in an embarrassingly parallel manner and allowing gradient updates (OPT) to be performed.

**Restrict/Prolongate blocks:** In the EM algorithm we follow Trask et al. (2022) where the standard evidence lower bound derivation of expectation maximization yields closed form expressions for the posterior distribution of the gating function conditioned on the data. For coarse and fine scales, we obtain the expressions

$$w_{id} := p(Z_1 = i | Y_1 = y_d) = \frac{\pi_i(\boldsymbol{x}_d, \theta)\mathcal{N}(y_d; \mu_i(\boldsymbol{x}_d), \sigma_i^2\mathbf{I})}{\sum_I \pi_I(\boldsymbol{x}_d)\mathcal{N}(y_d; \mu_I(\boldsymbol{x}_d), \sigma_I^2\mathbf{I})}, \quad (4)$$

$$w_{ijd} := p(Z_1 = i, Z_2 = j | Y_2 = y_d) = \frac{\pi_i(\boldsymbol{x}_d, \theta)\pi_{ij}(\boldsymbol{x}, \theta_i)\mathcal{N}(y_d; \mu_{ij}(\boldsymbol{x}_d), \sigma_{ij}^2\mathbf{I})}{\sum_{I,J} \pi_I(\boldsymbol{x}_d)\pi_{IJ}(\boldsymbol{x}, \theta_I)\mathcal{N}(y_d; \mu_{IJ}(\boldsymbol{x}_d), \sigma_{IJ}^2\mathbf{I})}, \quad (5)$$

We note that the coarse posterior $w_{id}$ can be evaluated independently of the fine level, and then sequentially used to evaluate the fine level posterior $w_{ijd}$. Following the nomenclature in multigrid methods, we refer to this evaluation of $w_{ijd}$ from $w_{id}$ as a *restriction*. Then, to propagate information from the child to parent the following marginalization formula is used

$$\hat{w}_{id} := p(Z_1 = i | Y_2 = y_d) = \frac{\sum_j \pi_i(\boldsymbol{x}_d)\pi_{ij}(\boldsymbol{x}, \theta_i)\mathcal{N}(y_d; \mu_{ij}(\boldsymbol{x}_d), \sigma_{ij}^2\mathbf{I})}{\sum_{I,j} \pi_I(\boldsymbol{x}_d)\pi_{Ij}(\boldsymbol{x}, \theta_I)\mathcal{N}(y_d; \mu_{Ij}(\boldsymbol{x}_d), \sigma_{Ij}^2\mathbf{I})}. \quad (6)$$

Again, motivated multigrid nomenclature, we refer to this marginalization as *prolongation*.

**LSS block.** At each scale a posterior weighted least square solve prescribes optimal values for final linear layers of expert models. In Trask et al. (2022) these least squares problems are defined as:

$$\sum_{d=1}^{N_d} w_{id}\Phi_\alpha(\boldsymbol{x_d})\Phi_\beta(\boldsymbol{x_d})c_{i,\beta} = \sum_{d=1}^{N_d} w_{id}\Phi_\alpha(\boldsymbol{x_d})y_d, \quad (7)$$

$$\sum_{d=1}^{N_d} w_{ijd}\Phi_\alpha(\boldsymbol{x_d})\Phi_\beta(\boldsymbol{x_d})c_{ij,\beta} = \sum_{d=1}^{N_d} w_{ijd}\Phi_\alpha(\boldsymbol{x_d})y_d. \quad (8)$$

In the current work, we replace both with the hierarchical cooperative/competitive loss defined in LS.

**Coop/comp split block.** In the literature, the incorporation of optimal layers requires choosing between a dense solve in the LSS block for a purely cooperative formulation Lee et al. (2021), or a parallelizable but purely competitive loss Trask et al. (2022). Our primary contribution is a splitting scheme outlined in equation I which preserves the benefits of both. Analysis in Theorem 1 provides guaranteed conditions for convergence of this scheme, and experimental results in Figure 3 show that it exposes a source of GPU parallelism

**OPT block.** A standard gradient descent update is finally applied at each layer of the hierarchy. As explained in Trask et al. (2022), the loss may be derived rigorously from the evidence lower bound and results in a cross-entropy loss penalizing mismatch between the posterior distribution and gating functions. In light of policy gradient methods, one may interpret the posterior as a reward and the gating network as a policy. The updates are thus localized to each level, exposing the source of parallelism that JAX's `vmap` is able to exploit (See Figure 3).

$$\mathcal{L}_c(\theta; \mathcal{D}) = \sum_{i,d} w_{id} \log \pi_i(\boldsymbol{x_d}; \theta), \tag{9}$$

$$\mathcal{L}_f(\theta_i; \mathcal{D}) = \sum_{i,j,d} w_{ijd} \log \left( \pi_i(\boldsymbol{x_d}; \theta) \pi_{ij}(\boldsymbol{x_d}; \theta_i) \right), \tag{10}$$

$$\mathcal{L}_{f2c}(\theta; \mathcal{D}) = \sum_{i,d} \hat{w}_{id} \log \pi_i(\boldsymbol{x_d}; \theta). \tag{11}$$

A key equation from Trask et al. (2022) demonstrates that the total loss decouples across scales:

$$\mathcal{L}_f(\theta_i; \mathcal{D}) = \mathcal{L}_{f2c}(\theta; \mathcal{D}) + \sum_{i,j,d} w_{ijd} \log \left( \pi_{ij}(\boldsymbol{x_d}; \theta_i) \right). \tag{12}$$

This decoupling allows the scales to separate so that training at the fine level does not impede progress from the coarse. A useful interpretation of this is that the the prolongated loss at the coarse scale may be viewed as a correction to the fine scale.

# B   PROOF OF THEOREM 1

We first summarize why $\rho(M^{-1}N) < 1$ is a sufficient condition for convergence.

With the scheme given by

$$M c_{k+1} = N c_k + b, \tag{13}$$

we may rewrite to obtain

$$c_{k+1} = c_k + M^{-1} \left( N - M \right) c_k + M^{-1} b. \tag{14}$$

We now prove that $\rho(M^{-1}N) < 1$. For simplicity, we adopt the notation

$$N = n \, \mathbf{P}^\intercal \mathbf{A} \mathbf{P} \tag{15}$$

$$M = m \, \mathbf{P}^\intercal \mathbf{B} \mathbf{P} \tag{16}$$

where $m = \sigma_{comp}^{-2}, n = -\sigma_{coop}^{-2}$ and $m, n > 0$ denote the variance scaling, $\mathbf{P} \in \mathbb{R}^{\mathrm{N}_{width} \times \mathrm{N}_{experts} \times \mathrm{N}_{data}}$ denotes the output of the $i^{th}$ output neuron of the $j^{th}$ expert model evaluated at the $k^{th}$ node, and $\mathbf{A}$ and $\mathbf{B}$ denote a scaling by either the gating or posterior distribution, respectively.

$$\mathbf{A}_{ij} = \sum_d \pi_{id} \pi_{jd} \tag{17}$$

$$\mathbf{B}_{ij} = \sum_d w_{id} \delta_{ij}. \tag{18}$$

After an appropriate reshaping, it is clear that equation 15 are matrices associated with weighted least squares problems, with $\mathbf{A}$ and $\mathbf{B}$ serving as weights, and are symmetric positive definite provided the weights are non-degenerate.

Rewriting the desired spectral radius inequality in terms of equation 15 and lower bounding the spectral radius by the maximal generalized Rayleigh quotient, we obtain

$$\frac{\sigma_{coop}^2}{\sigma_{comp}^2} > \max_x \frac{x^\mathsf{T} \mathbf{P}^\mathsf{T} \mathbf{A} \mathbf{P} x}{x^\mathsf{T} \mathbf{P}^\mathsf{T} \mathbf{B} \mathbf{P} x}, \tag{19}$$

or after simplifying by defining $y = \mathbf{P}x$

$$\frac{\sigma_{coop}^2}{\sigma_{comp}^2} > \max_y \frac{y^\mathsf{T} \mathbf{A} y}{y^\mathsf{T} \mathbf{B} y}. \tag{20}$$

The numerator and denominator can be treated in worst case by bounding by the maximum and minimum eigenvalues, respectively

$$\frac{\sigma_{coop}^2}{\sigma_{comp}^2} \geq \frac{(\max \lambda_A)\, y^\mathsf{T} y}{(\min \lambda_B)\, y^\mathsf{T} y} = \frac{\max \lambda_A}{\min \lambda_B}. \tag{21}$$

As both matrices are positive definite, we can bound using standard element-wise expressions following the Gerschgorin circle theorem Gerschgorin (1931).

$$\max \lambda_A \geq \max_i \sum_j |A_{ij}| \tag{22}$$

$$\min \lambda_B \leq \min_i \left( |B_{ii}| - \sum_{j \neq i} |B_{ij}| \right) \tag{23}$$

$$\frac{\sigma_{coop}^2}{\sigma_{comp}^2} \geq \frac{\max\limits_i \sum\limits_j |A_{ij}|}{\min\limits_i \left( |B_{ii}| - \sum\limits_{j \neq i} |B_{ij}| \right)}. \tag{24}$$

By direct calculation we compute

$$\sum_j |A_{ij}| = \sum_{d,j} \pi_{id} \pi_{jd} \tag{25}$$

$$= \sum_d \pi_{id} \tag{26}$$

and

$$|B_{ii}| - \sum_{j \neq i} |B_{ij}| = \sum_d w_{id} \delta_{ij} - \sum_{d, j \neq i} w_{id} \delta_{ij} \tag{27}$$

$$= \sum_d w_{id} \tag{28}$$

Finally providing our desired bound

$$\frac{\sigma_{coop}^2}{\sigma_{comp}^2} \geq \frac{\max\limits_i \sum\limits_d \pi_{id}}{\min\limits_i \sum\limits_d w_{id}}. \tag{29}$$

**Validity of assumptions, anticipated consequences, and practical use.** In practice, there are several assumptions that may not be met. The expert models may not be linearly independent (particularly at initialization time); we explore this result experimentally in the Section 4.3. Secondly, in the event of expert posterior collapse (i.e. there exists an $i$ where $w_{id} = 0$ for all $d$) the estimate could give a division by zero. In exact precision, $w_{id}$ is never zero, as it is the posterior distribution of a Gaussian mixture and Gaussians have non-compact support. In practice, in the extreme tails of the Gaussian contributions the denominator may be vanishingly small, requiring a small stabilizing background white noise in the mixture model to avoid division by zero in machine precision.

## C    DERIVATIONS FOR PINN APPLICATION

For a general PDE of the form

$$
\begin{aligned}
\mathcal{L}[u](\boldsymbol{x}, t) &= f(\boldsymbol{x}, t), \quad \boldsymbol{x} \in \Omega \\
\mathcal{B}[u](\boldsymbol{x}, t) &= g(\boldsymbol{x}, t), \quad \boldsymbol{x} \in \partial\Omega \\
u(\boldsymbol{x}, 0) &= u_0(\boldsymbol{x}), \quad t = 0
\end{aligned}
$$

where $\mathcal{L}$ is any derivative operator, we can modify equation equation 1 with the residue in place of the true prediction and add the terms for boundary/initial conditions to the mean-field approximation. For simplicity, we only show the boundary condition terms, as the initial condition terms are analogous

$$
p(\mathcal{E}_{i_1} = u | Z_1 = i_1) = \frac{1}{Q_{i_1}} \mathcal{N}(f; \mathcal{L}[\mu_{i_1}], \sigma_{r_1}^2) \mathcal{N}(f; \mathcal{L}[\hat{y}_1], \sigma_{r_2}^2) \mathcal{N}(\gamma g; \gamma \mu_{i_1}, \sigma_{b_1}^2) \mathcal{N}(\gamma g; \gamma \hat{y}_1, \sigma_{b_2}^2)
$$

where $\gamma$ is an indicator function, i.e.

$$
\gamma(\boldsymbol{x}) = \begin{cases} 1 & \text{if } \boldsymbol{x} \in \partial\Omega \\ 0 & \text{otherwise} \end{cases}
$$

The observed data log-likelihood becomes

$$
\log L = \sum_{d=1}^{N_d} \log \left[ \sum_{I_n} \tilde{\boldsymbol{\pi}}_{I_n}^d \mathcal{N}(f^d; \mathcal{L}[\mu_{I_n}]^d, \sigma_{r_1}^2) \mathcal{N}(f^d; \mathcal{L}[\hat{y}_n]^d, \sigma_{r_2}^2) \mathcal{N}(\gamma^d g^d; \gamma^d \mu_{I_n}^d, \sigma_{b_1}^2) \mathcal{N}(\gamma^d g^d; \gamma^d \hat{y}_n^d, \sigma_{b_2}^2) \right]
$$

from which a new ELBO is given in a similar manner to the regression problem. In the E-step, we compute the posterior distribution, i.e.,

$$
w_{I_n}(\boldsymbol{x}^d) = \frac{\tilde{\boldsymbol{\pi}}_{I_n} \mathcal{N}(f^d; \mathcal{L}[\mu_{I_n}](\boldsymbol{x}^d), \sigma_{r_1}^2) \mathcal{N}(\gamma g^d; \gamma \mu_{I_n}^d, \sigma_{b_1}^2)}{\sum_{J_n} \tilde{\boldsymbol{\pi}}_{J_n} \mathcal{N}(f^d; \mathcal{L}[\mu_{J_n}](\boldsymbol{x}^d), \sigma_{r_1}^2) \mathcal{N}(\gamma g^d; \gamma \mu_{I_n}^d, \sigma_{b_1}^2)}.
$$

If $\mathcal{L}$ is a linear operator, taking the derivative of the ELBO with respect to the expert coefficients yields a similar weighted least-squares problem to equation equation LS, where the matrix on the left-hand side

$$
\sum_d \left[ \frac{w_{I_n}^d \delta_{I_n J_n}}{\sigma_{r_1}^2} \mathcal{L}[H]_{I_n, \alpha}^d \mathcal{L}[H]_{J_n, \beta}^d + \frac{\mathcal{L}[\tilde{\boldsymbol{\pi}}_{I_n} H_{I_n, \alpha}]^d \mathcal{L}[\tilde{\boldsymbol{\pi}}_{J_n} H_{J_n, \beta}]^d}{\sigma_{r_2}^2} + \gamma^d \left( \frac{w_{I_n}^d \delta_{I_n J_n}}{\sigma_{b_1}^2} + \frac{\tilde{\boldsymbol{\pi}}_{I_n}^d \tilde{\boldsymbol{\pi}}_{J_n}^d}{\sigma_{b_2}^2} \right) H_{I_n, \alpha}^d H_{J_n, \beta}^d \right]
$$

and the right-hand side vector

$$
\sum_d \left[ \left( \frac{w_{I_n}^d \mathcal{L}[H]_{I_n, \alpha}^d}{\sigma_{r_1}^2} + \frac{\mathcal{L}[\tilde{\boldsymbol{\pi}}_{I_n} H_{I_n, \alpha}]^d}{\sigma_{r_2}^2} \right) f^d + \gamma^d \left( \frac{w_{I_n}^d}{\sigma_{b_1}^2} + \frac{\tilde{\boldsymbol{\pi}}_{I_n}^d}{\sigma_{b_2}^2} \right) g^d H_{I_n, \alpha}^d \right]
$$

If $\mathcal{L}$ is nonlinear, we will need to set up a Newton solver. $\boldsymbol{H}_{I_n}$ and $\tilde{\boldsymbol{\pi}}_{I_n}$ are optimized with a gradient descent loss

$$
\mathcal{L}_{\text{GD}} = -\ell(\theta)
$$

similar to the regression problem.

## D    EXPERIMENTAL DETAILS AND HYPERPARAMETERS

Simulations were conducted on an Nvidia A100 cluster, an Nvidia H200 cluster, as well initial experiments prototyped on a Macbook Air M2 16GB/256GB. Scripts for experiments will be provided on the anonymized github with associated seeds.

### D.1    HYPERPARAMETERS USED FOR DATA COLLECTION

**Remark.** For the PINN case the additional terms present in the multiobjective loss require special care. To accelerate hyperparameter tuning for the ablation study of Peclet number the splitting scheme was turned off and the SOR scheme turned on with small $\omega$ for a few cases to ensure convergence.

Table 1: Key hyperparameters used in section 4.1.

| Component | Parameter | Value |
|---|---|---|
| **Problem** | Dataset | 1D Gaussian peak, $\mu = 0.5$, $\sigma = 0.05$ |
| | Input dimension | 1 |
| **Gating Network** | Hidden units / Depth (excluding input layer) | 30 / 0 |
| | Activation | `tanh` |
| **Hierarchical MoE** | Partitions ($N_c$, $N_f$) | 3, 2 |
| | Basis size / Hidden units / Depth | 30 / 30 / 1 |
| **Training** | Outer iterations | 50,000 (staged training) |
| | Learning rates (coarse/fine) | 1e-3 / 1e-3 |
| | $\sigma_{\text{comp}}$ / $\sigma_{\text{coop}}$ | 1.0 / 1e6 |
| **Iterative Solver** | Type | Iterative splitting scheme |
| | Tolerance / Regularization | 1e-12 / 1e-4 |
| | Max iterations | 10,000 |
| **Precision** | Data and parameters | `float64` |

Table 2: Key hyperparameters used for strong scaling study (section 4.2).

| Component | Parameter | Value |
|---|---|---|
| **Problem** | Dataset | Gaussian peak, $\mu = 0.5$, $\sigma = 0.1$ |
| | Input dimension | 64 |
| **Gating Network** | Hidden units / Depth (excluding input layer) | 30 / 0 |
| | Activation | `tanh` |
| **Hierarchical MoE** | Partitions sweep ($N_c$, $N_f$) | $(4, 2), (4, 4), (4, 8), (4, 16), (4, 32)$ |
| | Basis size / Hidden units / Depth | 30 / 20 / 1 |
| **Training** | Benchmark iterations | 5000 (solver timing only) |
| | Learning rates (coarse/fine) | 1e-3 / 1e-3 |
| | $\sigma_{\text{comp}}$ / $\sigma_{\text{coop}}$ | 1.0 / 1e5 |
| **Iterative Solver** | Type | Direct / Iterative splitting scheme (with and without `vmap`) |
| | Tolerance / Regularization | 1e-12 / 1e-4 |
| | Max iterations | 5000 |
| **Precision** | Data and parameters | `float32` |

Table 3: Key hyperparameters used in section 4.3.

| Component | Parameter | Value |
|---|---|---|
| **Problem** | Dataset | 1D Gaussian peak, $\mu = 0.5$, $\sigma = 0.05$ |
| | Input dimension | 1 |
| **Gating Network** | Hidden units / Depth (excluding input layer) | 50 / 0 |
| | Activation | `tanh` |
| **Hierarchical MoE** | Partitions ($N_c$, $N_f$) | 4, 2 |
| | Basis size / Hidden units / Depth | 20 / 20 / 1 |
| **Training** | Outer iterations | 5,000 |
| | Learning rates (coarse/fine) | 1e-3 / 1e-3 |
| | $\sigma_{\text{comp}}$ / $\sigma_{\text{coop}}$ | 1.0 / 1e5 |
| **Iterative Solver** | Type | Iterative splitting scheme |
| | Tolerance / Regularization | 1e-12 / 1e-4 |
| | Max iterations | 2,000 |
| | Recorded iterations counts | performed for $\sigma_{\text{coop}}$ given in legend |
| **Precision** | Data and parameters | `float64` |

Table 4: Key hyperparameters used for linear solve iteration scaling with problem dimension (section 4.4).

| Component | Parameter | Value |
|---|---|---|
| **Problem** | Dataset | 1D Gaussian peak, $\mu = 0.5$, $\sigma = 0.1$ |
| | Input dimension | 1, 2, 4, 8, 16, 32, 64, 128 |
| **Gating Network** | Hidden units / Depth (excluding input layer) | 100 / 0 |
| | Activation | `tanh` |
| **Hierarchical MoE** | Coarse / Fine partitions ($N_c$, $N_f$) | 4 / 2 |
| | Basis size / Hidden units / Depth | 10 / 10 / 1 |
| **Training** | Outer iterations | 100,000 |
| | Accuracy threshold | 1% relative error |
| | Learning rates (coarse/fine) | 1e-3 / 1e-3 |
| | $\sigma_{\text{comp}}$ / $\sigma_{\text{coop}}$ | 1.0 / 1e5 |
| **Iterative Solver** | Type | Iterative splitting scheme |
| | Tolerance / Regularization | 1e-12 / 1e-4 |
| | Max iterations | 100,000 |
| **Precision** | Data and parameters | `float64` |

Table 5: Key hyperparameters used for convection-diffusion problem (section 4.5).

| Component | Parameter | Value |
|---|---|---|
| **Problem** | Dataset | 100x100 grid points |
| **Gating Network** | Hidden units / Depth (excluding input layer) | 40/0 or 20/0 |
| | Activation | `tanh` |
| **Hierarchical MoE** | Coarse / Fine partitions ($N_c$, $N_f$) | 4 / 2 |
| | Basis size / Hidden units / Depth | MoEs: 10 / 10 / 2 PINNs: 40/40/3 |
| **Training** | Outer iterations | MoEs: 50,000 PINNs: 100,000 |
| | Learning rate | 1e-4 |
| | $\sigma_{\text{comp}}$ / $\sigma_{\text{coop}}$ | 1e-3 / 1e-3 |
| **Precision** | Data and parameters | `float64` |

Table 6: Key hyperparameters used for Peclet number ablation study (section 4.5).

| Component | Parameter | Value |
|---|---|---|
| **Problem** | Dataset | 500 points on [0,1] |
| **Gating Network** | Hidden units / Depth (excluding input layer) | 40/1 |
| | Activation | `tanh` |
| **Hierarchical MoE** | $N_c$ (1 level) | 4 |
| | Basis size / Hidden units / Depth | MoEs: 10 / 10 / 0 PINNs: 40/30/4 |
| **Training** | Outer iterations | MoEs: 30,000 PINNs: 100,000 |
| | Learning rate | 1e-4 |
| | $\sigma_{\text{comp}}$ / $\sigma_{\text{coop}}$ | 1e5 / 1e-3 |
| **Precision** | Data and parameters | `float64` |

