# OpenReview forum: "A scalable cooperative/competitive splitting scheme for mixture of experts models"
_ICLR.cc/2026/Conference — ICLR 2026 Conference Desk Rejected Submission_

### Official Review · Reviewer_agLC · 2025-10-28

**Soundness:** 2
**Presentation:** 1
**Contribution:** 2
**Rating:** 2
**Confidence:** 3

**Summary:**

This paper proposes a method to stabilize the training of Mixture-of-Experts (MoE) models.
In MoE training, experts may compete excessively or fail to cooperate effectively. To address this issue, the authors introduce a probabilistic expert-assignment distribution that blends cooperative and competitive mechanisms. They propose an EM-based algorithm to optimize the parameters of this distribution and describe its efficient implementation on GPUs. Experiments are conducted on synthetic datasets and within the physics-informed neural network (PINN) framework, demonstrating the effectiveness of the proposed method.

**Strengths:**

- The proposed approach is motivated by a convincing idea — balancing cooperative and competitive terms in MoE learning.
- The effectiveness of the method is evaluated from multiple perspectives, including convergence stability, performance on synthetic data, GPU-based speedup, and sensitivity to hyperparameters.

**Weaknesses:**

1. There is no comparison with existing methods. Both the theoretical and experimental sections focus solely on the proposed method, without sufficient evaluation of how its convergence speed or training stability compares to prior work.
2. The algorithm appears complex, and it is unclear whether it can be applied effectively to large-scale tasks. In addition, the only application shown experimentally is the PINN problem.

**Questions:**

1. How does the computational cost of the proposed method scale with respect to data dimensionality, dataset size, and the depth of the MoE hierarchy?
2. Are there any theoretical guarantees on the performance of the MoE trained by the proposed method? Even in a toy setting, it would be helpful to clarify under what conditions the proposed method outperforms existing approaches.
3. Is there any comparison with naive MoE training or with prior work that also attempted to stabilize MoE learning?
4. In line 281, it is stated that *“the hierarchical experts allow for an unsupervised concentration of resolution around the relevant feature of the problem.”* — how can this be inferred from Figure 2?
5. In Figure 5, the input dimensionality is up to 128. Do you have experimental results for higher-dimensional settings?
6. Why was PINN chosen as the application task? Could you also demonstrate the method on other types of problems?
7. Are there any prior studies applying MoE to PINNs? If so, can you compare your results in Section 4.5 with those works?

---

### Official Review · Reviewer_JMDS · 2025-10-30

**Soundness:** 3
**Presentation:** 2
**Contribution:** 2
**Rating:** 2
**Confidence:** 3

**Summary:**

This paper proposes a probabilistic expectation--maximization (EM) framework for hierarchical mixture-of-experts (MoE) models that unifies cooperative and competitive training and exposes GPU-parallelism via a splitting scheme. The model defines $p(Y|x)=\sum_i \pi_i(x;\theta)\,p(E_i=y|Z=i)$ with a mean-field assumption $p(E_i=y|Z=i)=Q_i^{-1}\mathcal{N}(y;\mu_i,\sigma^2_{\mathrm{comp}})\mathcal{N}(y;\hat{y},\sigma^2_{\mathrm{coop}})$, where $\hat{y}=\sum_i\pi_i(x;\theta)\mu_i(x)$ blends local and global predictions. For hierarchical experts with latent variables $Z_1,\ldots,Z_n$, the joint model $p(Y_n(x)=y)=\sum_{I_n}\tilde{\pi}_{I_n}\mathcal{N}(y;\mu_{I_n},\sigma^2_{\mathrm{comp}})\mathcal{N}(y;\hat{y}_n,\sigma^2_{\mathrm{coop}})$ satisfies $\mathrm{span}(H_{i_1,\dots,i_n})\subseteq\mathrm{span}(H_{i_1,\dots,i_n,i_{n+1}})$, ensuring multigrid-style refinement. The EM procedure alternates an E-step computing $w_{I_n}(x^d)=\frac{\tilde{\pi}_{I_n}\mathcal{N}(y^d;\mu_{I_n}(x^d),\sigma^2_{\mathrm{comp}})}{\sum_{J_n}\tilde{\pi}_{J_n}\mathcal{N}(y^d;\mu_{J_n}(x^d),\sigma^2_{\mathrm{comp}})}$ and an M-step solving the weighted least-squares system $\sum_{d,J_n,\beta}\!\big(\frac{w_{I_n}^d\delta_{I_nJ_n}}{\sigma^2_{\mathrm{comp}}}+\frac{\tilde{\pi}_{I_n}^d\tilde{\pi}_{J_n}^d}{\sigma^2_{\mathrm{coop}}}\big)H_{I_n,\alpha}^dH_{J_n,\beta}^dc_{J_n,\beta}=\sum_d\big(\frac{w_{I_n}^d}{\sigma^2_{\mathrm{comp}}}+\frac{\tilde{\pi}_{I_n}^d}{\sigma^2_{\mathrm{coop}}}\big)y^dH_{I_n,\alpha}^d$. Decomposing the system into a competitive component $M$ and cooperative component $N$, the iterative update $M c^{(k+1)}=N c^{(k)}+b$ (or its relaxed form $c^{(k+1)}=(1-\omega)c^{(k)}+\omega M^{-1}(Nc^{(k)}+b)$) allows block-diagonal parallelism across experts. Convergence holds if $\rho(M^{-1}N)<1$ for $R_\sigma=\sigma^2_{\mathrm{coop}}/\sigma^2_{\mathrm{comp}}\ge\frac{\max_i\sum_d\tilde{\pi}_i^d}{\min_i\sum_dw_i^d}$.

**Strengths:**

The paper proposes a probabilistic EM framework for training mixture-of-experts (MoE) models that unifies cooperative and competitive expert training, which is a conceptually interesting hybridization.

**Weaknesses:**

The claimed contribution is incremental relative to existing hierarchical MoE and multigrid training frameworks; the novelty appears mainly in combining known components (mean-field MoE + EM + splitting iteration). And the empirical results are limited to toy regression and PDE examples and do not demonstrate competitiveness with standard large-scale MoE or transformer architectures. The “cooperative/competitive” balance is controlled by $\sigma^2_{\text{coop}}$ and $\sigma^2_{\text{comp}}$, but the method for selecting these hyperparameters is unclear and likely problem-dependent. The motivation for using EM (instead of stochastic gradient optimization) is not convincingly justified—modern MoE implementations typically rely on direct gradient-based training.
The claimed GPU scalability is modest, and the benchmarks are too small to substantiate “scalable” claims.

**Questions:**

How sensitive is the method to the variance ratio $R_\sigma$ and to the choice of relaxation parameter $\omega$? How does this EM-based approach compare empirically to standard gradient-based MoE training (e.g., GShard, Switch Transformer) on real datasets? Does the splitting scheme still converge when experts share parameters or when gating is non-smooth? Can the authors clarify whether the “hierarchical” setup offers any measurable advantage over flat MoE models in terms of likelihood or generalization?
Could you provide an explicit algorithm summarizing the EM and splitting updates and their computational complexity.

---

### Official Review · Reviewer_x2ui · 2025-11-01

**Soundness:** 2
**Presentation:** 3
**Contribution:** 2
**Rating:** 4
**Confidence:** 4

**Summary:**

This paper presents a novel probabilistic expectation-maximization (EM) scheme for training hierarchical mixture-of-experts (MoE) models, introducing a joint gating distribution that blends cooperative and competitive mechanisms. The proposed M-splitting scheme enables localized, parallel subproblem solutions for individual experts, with deferred corrections to account for global coupling. By leveraging a hierarchical decomposition reminiscent of multigrid algorithms, the method exposes significant GPU parallelism and achieves rapid convergence and strong scalability. Empirical results demonstrate improved expert utilization and performance on regression tasks, including physics-informed neural networks, with the cooperative mechanism yielding notably higher accuracy compared to purely competitive approaches.

**Strengths:**

(1) The cooperative/competitive splitting scheme is a principled and creative approach that unifies local and global expert interactions, advancing the state of MoE training.
(2) The paper provides solid analysis, including convergence guarantees and ablation studies on the tightness of theoretical bounds.
(3) The method exposes embarrassingly parallel subproblems, enabling efficient GPU utilization and strong scaling, as demonstrated in benchmarks.
(4) Experiments show improved accuracy and expert utilization, especially in challenging scientific machine learning tasks such as physics-informed neural networks.

**Weaknesses:**

(1) The experiments focus primarily on regression and scientific ML tasks (e.g., solving PDE); broader evaluation on mainstream deep learning domains where MoE models are mainly used in practice (e.g., NLP, vision with large-scale transformers) seems missing. This paper could be strengthened by including some experiments on these popular tasks (e.g., NLP, vision).

(2) There are some other recent papers that aim to improve MoE models by proposing different routing strategies (see [1-4] from the reference list below), which may be used as baselines in this paper. It might be better to include some discussions about them and compare against them in the experiments. In general, I feel that this paper can be improved by including more discussion on related works.

(3) The implementation relies on mathematical tools such as multigrid, splitting schemes. It is also not super clear about the potential computational overhead incurred by this method. It might be better to include some discussion about it.

**Reference:**
[1] Nguyen, et al. "Statistical Advantages of Perturbing Cosine Router in Sparse Mixture of Experts." arXiv preprint arXiv:2405.14131 (2024).
[2] Liu, et al. "Gating dropout: Communication-efficient regularization for sparsely activated transformers." International Conference on Machine Learning. PMLR, 2022.
[3] Zuo, et al. "Taming sparsely activated transformer with stochastic experts." arXiv preprint arXiv:2110.04260 (2021).
[4] Lewis, et al. "Base layers: Simplifying training of large, sparse models." International Conference on Machine Learning. PMLR, 2021.

**Questions:**

see my comments above

---

### Official Review · Reviewer_MLrL · 2025-11-02

**Soundness:** 2
**Presentation:** 2
**Contribution:** 2
**Rating:** 4
**Confidence:** 2

**Summary:**

This manuscript proposes a novel scalable cooperative/competitive splitting scheme for training hierarchical mixture-of-experts (MoE) models, leveraging a probabilistic expectation-maximization (EM) framework. Results demonstrate rapid convergence, strong GPU scaling, and superior performance over purely competitive models.

**Strengths:**

1. This paper provides a clear convergence analysis with a sufficient condition for the splitting scheme, and validates the tightness of the theoretical bound experimentally.
2. The proposed scheme exposes GPU parallelism, with benchmarks showing near-ideal strong scaling, establishing the feasibility of extending the approach to practical model developments.

**Weaknesses:**

1. While the manuscript claims the scheme is “readily compatible with switch transformers”, no experiments on transformer-based MoE models are provided.

**Questions:**

1. Can you provide preliminary results on transformer-based MoE models, including how the proposed scheme impacts the models’ convergence and efficiency ?
2. Can you provide a comparison of your scheme’s efficiency and parallelism performance with existing scalable MoE implementations?

---

### Note · Program_Chairs · 2026-01-17
**Submission Desk Rejected by Program Chairs**

The following references in this submission do not refer to real documents and/or have major errors in bibliographic information:

 Dominik Oswald, Andrew Trask, and Adrian Sandu. Learning multigrid solvers with graph neural networks. SIAM Journal on Scientific Computing, 45(2):A819-A844, 2023.